

# Application of the Governance Disruptions Framework to German agricultural soil policy

Bartosz Bartkowski[1], Stephan Bartke[1], Nina Hagemann[2], Bernd Hansjürgens[1], Christoph Schröter-Schlaack[1]

[1]UFZ – Helmholtz Centre for Environmental Research, Department of Economics, Leipzig, Germany
[2]TU Dresden, International Institute Zittau, Zittau, Germany

*Correspondence to*: Bartosz Bartkowski (bartosz.bartkowski@ufz.de)

**Abstract.** Governance of natural resources is inherently complex and requires navigating trade-offs at multiple dimensions.
In this paper, we present and operationalize the Governance Disruptions Framework (GDF) as a tool for holistic analysis of
natural resource governance systems. For each of the four dimensions of the framework (target adequacy, object adequacy,
instrument adequacy, and behavioural adequacy) we formulate guiding questions, to be used when applying the framework to
particular governance systems. We then demonstrate the use of GDF by applying it to the core of German agricultural soil
policy. We show that for each framework dimension, the governance system exhibits deficits, particularly with respect to
object adequacy and instrument adequacy. Furthermore, we use the GDF-based analysis to highlight research gaps. We find
that stakeholder analyses are a central gap across GDF dimensions.

## 1 Introduction

Governance of natural resources is an inherently complex challenge. It involves crafting, sustaining and transforming
institutions (formal and informal rules) to navigate conflicts and trade-offs with respect to property rights (Bartkowski et al.,
2018; Schlager and Ostrom, 1992), societal targets (Pradhan et al., 2017), preferences of various stakeholder groups (Cavender-
Bares et al., 2015). A broad, holistic view on governance arrangements is required in order to identify ways to make such
arrangements more effective in protecting critical natural resources.

A critical natural resource that is only slowly gaining attention and prominence is soil, especially agricultural soil (Keesstra et
al., 2016; Vogel et al., 2018). Despite being affected by a multitude of policies, land and soil degradation is an ongoing problem
globally (IPBES, 2018a) and in Europe (IPBES, 2018b; Panagos et al., 2018). In the European Union (EU), the proposal for a
Soil Framework Directive was rejected in 2006 due to claimed sufficient cover of soil protection in existing legal frameworks
of the EU and its member states (Glæsner et al., 2014).

The question whether soils are sufficiently protected by current policies in the EU has been addressed from multiple
perspectives, often focusing primarily on the identification of relevant policies (e.g. Juerges and Hansjürgens, 2018; Ronchi et
al., 2019; Stankovics et al., 2018; Turpin et al., 2017). Less frequently, authors tried to link these policies to soil threats, soil
functions (Glæsner et al., 2014), multifunctionality (Vrebos et al., 2017) or resilience (Juerges et al., 2018). These analyses
collectively suggest that soil protection through EU and member states remains deficient. The focus of this literature is,



however, on the relationship between a broad set of policy instruments and a narrow subset of soil-related policy objectives. However, a broader view of the multiple dimensions of agricultural soil governance is missing. This broader view, aiming to overcome the existing deficits in soil governance, can result from investigating basic assumptions and questions related to key

dimensions constituting the current governance arrangements.

In this article, we seek to analyse agricultural soil governance in a systematic manner: first, we present the Governance Disruptions Framework (GDF), a holistic conceptual framework for analysis of natural resource governance, originally proposed by Schröter-Schlaack and Hansjürgens (2019). We operationalize the framework by formulating questions for each of its dimensions. Second, we apply the framework in an exploratory way to offer a broad perspective on the shortcomings of

current governance of agricultural soils in Germany. We focus on the formulation and implementation of national and EU policies with relevance to agricultural soil protection. Using the previously formulated questions for each GDF dimension, we draw from available scientific literature to provide responses to these questions. Our overall aim is to identify gaps or, framed more positively, windows of opportunity in the German agricultural soil governance system and the interactions between these gaps/opportunities. Furthermore, we suggest areas particularly requiring further research based on our analysis.

## 45   2 Conceptual framework: governance disruptions

To properly analyse and navigate complex governance systems and the trade-offs they imply, there is a need for conceptual perspectives that can reflect this complexity. In this article, we adopt the GDF, which was originally proposed with the aim to overcome the shortcomings of simplistic governance models (Schröter-Schlaack and Hansjürgens, 2019).



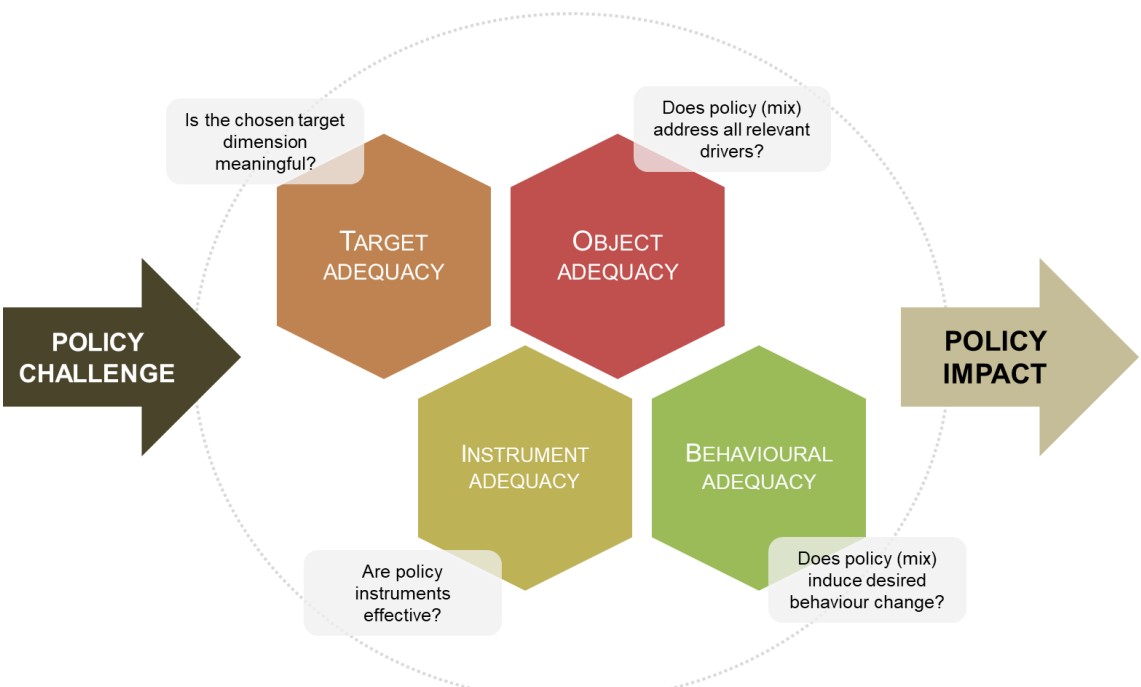

**Figure 1: Governance disruptions framework (GDF) (modified after Schröter-Schlaack and Hansjürgens, 2019, Fig. 48.2)**

The framework (Figure 1) consists of four interrelated elements that address different dimensions of a potential policy success/failure: target adequacy, object adequacy, instrument adequacy and behavioural adequacy. Together, they determine the impact of a governance arrangement or policy mix with respect to an underlying societal challenge, such as soil degradation. Disruptions in each dimension can lead to policy failure; interactions between the dimensions can have compound effects. Therefore, the benefit is in looking at all four of them together. In order to operationalize the framework, we formulate guiding questions for each of the dimensions, which we later apply to the context of agricultural soil governance in Germany.

**2.1 Target adequacy – are targets properly defined and meaningful?**

The first element of the disruptions framework is related to the identification of (an) adequate target dimension(s) to form the basis for policy intervention. For instance, EU agricultural targets conflict each other (e.g. farmers' income support, affordable food, environmental protection) (Pe'er et al., 2019) and are imprecise (e.g. referring to the ambiguous concept of 'good agricultural practice' as a baseline for the allocation of compliance costs between the farmer and the society) (Möckel, 2015a). Furthermore, often they target particular management practices (action-based approaches), rather than targeting changes in environmental objectives more directly (result-based approaches) (Burton and Schwarz, 2013). Given the heterogeneity of local environmental conditions, such contradicting and/or imprecise targets can result in ineffective or inefficient policies. Furthermore, understanding the relationships (synergies and trade-offs) between different target dimensions (e.g. different sustainability goals associated with different soil functions or ecosystem services) is crucial for a policy to be successful



(Schaafsma and Bartkowski, 2020). Tangible, clearly defined and measurable target dimensions are required for policy interventions – it needs to be clear what the target status is of, e.g., a multifunctional landscape, which requires a good understanding of both the biophysical potential of ecosystems (Seppelt et al., 2013) and the relevant societal preferences as

well as the identity and interests of those affected (Cavender-Bares et al., 2015). Failure to identify the target dimensions and properly understand the relationships among them, e.g. in agricultural policies, may make policy interventions ineffective at best, potentially aggravating soil degradation at worst. In other words, coherence is required already at the level of policy targets (Nilsson et al., 2012).

We propose the following questions to guide the analysis of the target adequacy of a policy mix:

Q1: What are the specific environmental objectives to be achieved? Which objectives can be clearly linked to concrete indicators that allow operationalization and monitoring?

Q2: Which spatial and temporal scales are relevant for each objective?

Q3: What interactions between individual objectives/indicators are known?

Q4: What are relevant stakeholder groups and their preferences towards the natural resource in question, and are there trade-
offs between the preferences of the stakeholder groups?

## 2.2 Object adequacy – are relevant (direct and indirect) drivers addressed?

The second element of the GDF relates to the identification of the causes (drivers[1]) of environmental degradation, i.e. the objects of policy intervention. Effective policy responses for more sustainable natural resource and ecosystem management and an effective navigation of trade-offs require identifying and addressing *all* relevant direct and indirect drivers of
environmental degradation. Policy interventions targeting only a specific driver or being restricted to a specific domain or sector (e.g. agricultural practices) may be ineffective if activities contributing to environmental degradation in other domains (e.g. companies' behaviour in the value chain or food consumption patterns) are not addressed as well. Domains include both human activities and natural processes (e.g. climate change and its effects on other environmental objectives). In the case of anthropogenic drivers, interests of stakeholder groups and power relationships become relevant (Berbés-Blázquez et al., 2016).
The necessity of identifying indirect drivers of environmental change has also been emphasised by the recent IPBES (2019) report on biodiversity and ecosystem services. Furthermore, it is crucial to take into account interactions between drivers (e.g. climate change increasing soil erosion) and between the policy interventions addressing them (e.g. subsidies for bioenergy production to fight climate change contradicting other environmental policies). Lastly, it is important to note that the role of particular drivers is usually not certain, especially in a particular local context.
The following questions may guide analyses of the object adequacy of a policy mix:

---

[1] In the context of object adequacy, both drivers and pressures in the sense of the driver–pressure–state–impact–response (DPSIR) framework are relevant. In the wording of GDF, DPSIR-pressures are direct, while DPSIR-drivers are indirect drivers of natural resource degradation.



Q1: Which (direct and indirect) drivers are known and what are the associated uncertainties?

Q2: How do the effects of the drivers vary across spatial and temporal scales?

Q3: Which drivers are particularly relevant/have particularly strong effects (and thus may require prioritization)?

Q4: What are known interactions among drivers?

Q5: Which stakeholder groups have interests associated with particular drivers (*qui bono?*)?

## 2.3 Instrument adequacy – are instruments properly chosen and designed?

The third element of the GDF is related to the choice and design of policy instruments. It focuses on individual instruments as well as their joint role within an instrument mix (Ring and Schröter-Schlaack, 2015). Common instrument-specific evaluation criteria are effectiveness and efficiency (Hanley et al., 1999)[2], but other criteria such as the capacity to support resilience and legitimacy have been discussed (e.g. Juerges et al., 2018). The usual classification of policy instruments distinguishes regulatory command-and-control instruments (e.g. spatial planning, mandatory management requirements), incentive-based instruments (e.g. taxes, tradable permits, subsidies) and 'soft', informational instruments (e.g. nudges, labelling, public procurement) (Bemelmans-Videc et al., 1998), though in practice, hybrid instruments are often found (Blackstock et al., 2020). Instrument choice and design is an art: Each instrument class has relative advantages and disadvantages, which need to be analysed in a context-specific way and especially in relation to the societal targets and drivers of environmental degradation that they are supposed to address. According to the so-called Tinbergen rule, each objective (target) should be associated with an instrument (Braathen, 2007; Tinbergen, 1952), though interactions between various instruments are particularly relevant for assessing each instrument's effectiveness and efficiency (Blackstock et al., 2020; Schader et al., 2014). The choice of the instrument mix influences the distribution of benefits among stakeholder groups, which can be used as guide to select appropriate instruments (Pannell, 2008) and is related to the issues discussed with respect to object adequacy (*qui bono?*). Also, it is important to keep in mind that policy instruments never operate in isolation. Rather, they are embedded in an existing institutional and cultural context that may influence their effectiveness and efficiency, especially with respect to transaction costs (Bolognesi and Nahrath, 2020).

We propose the following questions to guide analyses of instrument adequacy:

Q1: Which instruments address the environmental objective? Are there interdependencies?

Q2: How is the correspondence between the instruments and known drivers?

Q3: Are the individual instruments effective and efficient?

Q4: Is the mix of instruments coherent? Are there overlaps or gaps?

---

[2] There exist many subcriteria for both effectiveness and efficiency, including environmental effectiveness and additionality as dimensions of effectiveness, and cost-effectiveness and dynamic efficiency as dimensions of efficiency.



## 2.4 Behavioural adequacy – (how) will instruments trigger desired behavioural changes?

The fourth element of the GDF relates to the implicit or explicit assumptions about human behaviour embodied in particular instruments: does people's behaviour follow the predicted patterns, i.e. can the policy intervention effectively trigger desired behaviour change? Ultimately, this has crucial consequences for the effectiveness and efficiency of policy instruments. Sometimes, policy addressees do not respond to an otherwise well-designed intervention, because the underlying assumptions regarding their behaviour are incorrect. This is a particular challenge for conservation and management of natural ecosystems,

where a broad range of motivations (instrumental, social, intrinsic) for action is involved (Bartkowski and Bartke, 2018; Dessart et al., 2019). The heterogeneous distribution of motivations in the target population and among relevant stakeholder groups should be taken into account when designing policy interventions (Braito et al., 2020). Also, legitimacy and acceptance of the policy intervention by the addressees is an important issue here (Vainio et al., 2019), which points to interactions between behavioural adequacy and object adequacy (*qui bono?*). These considerations are strongly instrument-dependent – for instance,

incentive-based and 'soft' instruments can be highly sensitive to behavioural factors, while regulatory instruments are less so. On the other hand, even for the latter, compliance levels may vary depending on the interactions between institutional context (e.g. monitoring) and behavioural factors. More generally, asymmetric information between policy makers and land users is a challenge in which behavioural, social and institutional factors play a large role. In many cases, combinations of instruments may help to address the behavioural shortcomings of individual ones – e.g. the combination of monetary incentives with the

provision of information necessary to realize the incentivized behaviour change (e.g. Blackstock et al., 2020).

Following questions may be used to guide the analysis of behavioural adequacy:

Q1: How well do the underlying behavioural assumptions of the instruments reflect actual behaviour of the target population?

Q2: How well do the instruments reflect the heterogeneity of behavioural characteristics of the target population?

Q3: Are there behavioural complementarities or contradictions between the instruments?

## 3 Case study: Agricultural soil governance in Germany

In what follows, we apply the GDF, as introduced above, to the governance of agricultural soils in Germany. In order to leverage the breadth of the framework and demonstrate its potential to illuminate governance and policy analysis, we will address the GDF-based guiding questions (summarized in Table 1 below) and point out knowledge gaps. We focus on policy

documents and instruments in Germany that *explicitly* address agricultural soils. We therefore abstract from the many more instruments that have a rather implicit or indirect yet non-negligible relevance for soil governance (e.g. Nitrate Directive, Water Framework Directive or spatial planning laws) in order to keep our analysis focused. Germany is a particularly interesting example, because it is one of the few countries with explicit legislation for soil protection (Juerges and Hansjürgens, 2018; Ronchi et al., 2019). The German Federal Soil Protection Act (Bundes-Bodenschutzgesetz, BBodSchG) came into force

in 1999; it defines a framework for soil protection in Germany, with a particular focus on contaminated soils and their



restoration. Section 17 is explicitly dealing with agricultural soil management. Within the European Common Agricultural Policy (CAP), the main instruments addressing soil management explicitly are the good agricultural and environmental condition (GAEC) guidelines (1st CAP pillar) and the agri-environment and climate measures (AECM) (2nd pillar). Both instruments have been implemented in German law through ordinances. In addition, other relevant policies are referred to, 160 especially in the context of target adequacy.

## 3.1 Approach and methods

Table 1 summarizes the questions for each GDF dimension that will be used to guide the analysis of agricultural soil governance in Germany. Our analysis is based on the scientific literature, which we contrast with relevant policy documents where appropriate. In cases where the questions cannot be answered on the basis of literature and document analysis, we 165 formulate research/knowledge gaps.

**Table 1 Research questions for the explorative application of the governance disruptions framework to agricultural soil governance in Germany**

| Dimension | GDF questions |
| --- | --- |
| Target adequacy | What are the specific environmental objectives? Can these be linked to concrete indicators? |
| | Which spatial and temporal scales are relevant for each objective? |
| | What interactions between individual objectives/indicators are known? |
| | What are relevant stakeholder groups and their preferences towards agricultural soils, and are there trade-offs between the preferences of the stakeholder groups? |
| Object adequacy | Which (direct and indirect) drivers are known and what are the associated uncertainties? |
| | How do the effects of the drivers vary across spatial and temporal scales? |
| | Which drivers are particularly relevant/have particularly strong effects (and thus may require prioritization)? |
| | What are known interactions among drivers? |
| | Which stakeholder groups have interests associated with particular drivers (*qui bono?*)? |
| Instrument adequacy | Which instruments address the environmental objective? Are there interdependencies? |
| | How is the correspondence between the instruments and known drivers? |
| | Are the individual instruments effective and efficient? |
| | Is the mix of instruments coherent? Are there overlaps or gaps? |
| Behavioural adequacy | How well do the underlying behavioural assumptions of the instruments reflect actual behaviour of the target population? |
| | How well do the instruments reflect the heterogeneity of behavioural characteristics of the target population? |
| | Are there behavioural complementarities or contradictions between the instruments? |

## 3.2 Results and discussion

In this section, we discuss each dimension of the Governance Disruptions Framework (GDF) by addressing the questions from Table 1.



### 3.2.1 Target adequacy

*Q1: What are the specific environmental objectives? Can these be linked to concrete indicators?*

As Möckel (2015b) points out, there are essentially two policy documents that formulate explicit, legally binding targets
regarding agricultural soil management in Germany: the Federal Soil Protection Act (BBodSchG) and the Good Agricultural
and Environmental Conditions (GAEC) guidelines within CAP direct payments Cross-Compliance rules.[3] The BBodSchG
stresses the need to 'sustain and restore soil functions' with a strong focus on soil contamination, which reflects the historic
implementation purpose of the act to prevent harm from contaminated land (particularly for humans). The soil functions
invoked in the BBodSchG include 'natural functions', 'archive' for natural and cultural history and 'use functions', which are
only limitedly related to the soil functions concepts common in scientific literature (e.g. Vogel et al., 2018). Both the
BBodSchG and soil-relevant GAEC guidelines emphasize erosion protection and the requirement to sustain the site-specific
soil organic carbon (SOC) content. Furthermore, the BBodSchG also addresses avoiding soil compaction and sustaining and
improving biological activity in soils. Thus, while the BBodSchG seems to combine soil functions and soil threats in its
formulation of targets, the GAEC guidelines addresses only a limited set of threats. No specific indicators are formulated in
the analysed policy documents.

At the EU level, the Farm to Fork Strategy (F2F) and the Biodiversity Strategy as parts of the European Green Deal are also
relevant recent strategic documents that include references to soil protection (Montanarella and Panagos, 2021).[4] The
Biodiversity Strategy explicitly mentions 'limiting soil sealing', bringing back 'at least 10% of agricultural area under high-
diversity landscape features' to, among others, limit soil erosion, and preventing soil degradation for its ecosystem services
('soil fertility, nutrient cycling and climate regulation') (EC, 2020). At the even higher international level, the Sustainable
Development Goals (SDGs) have been used as a set of policy targets – however, as pointed out by Keesstra et al. (2016), while
soils are relevant for multiple SDGs, they are rarely explicitly mentioned. Moreover, the official list of SDGs indicators does
not refer to soils at all.

*Q2: Which spatial and temporal scales are relevant for each objective?*

Given the low specificity of the soil-related objectives identified in the above-mentioned policy documents, scale
considerations can only limitedly be assessed. Some work has been done on the temporal and spatial dimensions and
mismatches associated with different soil-based ecosystem services (mentioned in the EU Biodiversity Strategy), which can
be linked to soil functions (BBodSchG) (Bartkowski et al., 2020). However, the consequences of these mismatches for
governance have not yet been spelled out and constitute a research gap. It seems undisputed that the short to mid-term AECM
are not adequately in line with natural soil processes (Juerges et al., 2018; Vogel et al., 2018).

*Q3: What interactions between individual objectives are known?*

---

[3] The BBodSchG can be found at https://www.gesetze-im-internet.de/bbodschg/, the GAEC at https://eur-lex.europa.eu/legal-content/en/ALL/?uri=CELEX:32013R1306 (Annex II) (both accessed on 2.11.2020).
[4] As mentioned by Montanarella and Panagos (2021), further soil-relevant strategies are scheduled for 2021, e.g. the Zero Pollution Action Plan for Air, Water and Soil and an update of the Soil Thematic Strategy.



Given the general formulation of most targets related to agricultural soil governance in Germany, it is difficult to assess trade-offs between them. The exception is the BBodSchG's reference to soil functions, which are known to involve trade-offs (Bartkowski et al., 2020; Paul and Helming, 2019), especially when considered in specific environmental and management contexts (Schröder et al., 2020). For instance, organic fertilization is desirable in terms of nutrient cycling, but may be detrimental to water quality because of the mismatch between N/P ratios in organic fertilizer and the demands of most crops (idib.). However, these trade-offs are not explicitly acknowledged in the BBodSchG. With respect to the GAEC, it has been repeatedly noted that the superordinate CAP is an incoherent combination of targets and instruments, where contradictory environmental and production-oriented targets and instruments exist alongside each other (Pe'er et al., 2019).

*Q4: What are relevant stakeholder groups and their preferences towards the natural resource in question, and are there trade-offs between the preferences of the stakeholder groups?*

A tentative stakeholder analysis for German agricultural soil governance has been conducted by Jürges (2016). Because of the centrality of soils to agricultural production and environmental protection, the identified spectrum of actors is fairly broad, ranging from environmental and agricultural research institutes through farmers' associations to government ministries and agencies. We are not aware of any comprehensive analysis of their role in relevant policy processes (see also Techen et al., 2020). Also, given the general paucity of preference analyses in the context of agricultural soils in Germany (Bartkowski et al., 2020), possible trade-offs and, more generally, the relevance of preference heterogeneity are unknown. Stakeholder analyses constitute a major gap in soil governance research in Germany.

### 3.2.2 Object adequacy

*Q1: Which (direct and indirect) drivers are known and what are the associated uncertainties?*

Techen and Helming (2017) provide a comprehensive list of drivers of soil management, categorizing them into three classes: socio-economic, biophysical and technological. As socio-economic drivers, they identify: consumer demand, factor costs, policies and farm(er)s' attributes. The latter two will be addressed in the discussions of the next two dimensions, so we exclude them from the discussion of object adequacy. As biophysical drivers, Techen and Helming (2017) list: soil degradation threats, land availability, climate change and resource scarcity. Among technological drivers, they find: research, biomass technology, ICT & robotics and other technology. For the purposes of the present analysis, the latter class can be summarized as 'research and technology', except for 'biomass technology', which we consider as predominantly another (industrial) demand factor, adding to the conventional demand for food and feed that Techen and Helming (2017) subsume as 'consumer demand'. Soil (degradation) threats have been used to identify sectors contributing to soil degradation, including, next to agriculture, urbanization (land take), industry (contamination) and nature protection (with mostly co-benefits for sustainable soil management) (Glæsner et al., 2014). Land availability is of course a driver of intensification, and thus can be related to soil threats. Relatedly, ownership status of agricultural land (specifically, tenure) has frequently been hypothesized to negatively influence the sustainability of soil management (e.g. Soule et al., 2000); however, recent empirical insights from Europe have questioned this hypothesis (Daedlow et al., 2018; Leonhardt et al., 2019).





Looking at the soil-relevant governance arrangements discussed here, it is striking that rules mostly address the production side, ignoring the influence of other parts of the food chain from field to consumers. The F2F Strategy may change this with its explicit inclusion of consumption and the overall food chain, but its impact strongly depends on how it will interact with the CAP.

*Q2: How do the effects of the drivers vary across spatial and temporal scales?*

As indicated above, there are discordances between natural soil processes taking place at different time and spatial scales and the potential of governance to provide sensitive steering measures. We are not aware of holistic, systematic analyses of spatial variations between drivers of agricultural soil degradation in Germany. Some information is implicit in comparing scenario development processes with different spatial scales – e.g., Shared Socio-economic Pathways for European agriculture and food systems (Mitter et al., 2020) emphasize different drivers of change than global or national scenarios. However, while we are

aware of ongoing scenario development with a specific focus on Germany, which would help in this context, there are no existing results to draw upon. With regard to temporal scales, foresight (Techen and Helming, 2017) and scenario analyses (Mitter et al., 2020) can provide expert- or model-based hints at possible variation with respect to different drivers. However, since (the pace of) technological developments and value changes are inherently difficult to predict, there are limits to such analyses.

*Q3: Which drivers are particularly relevant/have particularly strong effects (and thus may require prioritization)?*

Prioritization for policy purposes, including prioritization of drivers of soil degradation, is an inherently normative exercise, and different prioritizations will be arrived at depending on the chosen yardstick. Drivers of soil degradation and related pressures are well-known and obvious not only to experts. Such drivers are climate change, economic growth or lifestyle preferences often leading to similar pressures (see e.g. Hagemann et al., 2019), which makes prioritization difficult. From the

point of view of soil research, some prioritization was undertaken by Techen et al. (2020), who identified 'cross-cutting research challenges' related to various drivers of soil degradation and to different soil threats and functions. However, these research challenges are not necessarily correlated with the policy-relevance and effect strength of drivers. Here, again, dedicated research efforts are required.

*Q4: Are there known interactions among drivers?*

The most obvious source of interactions between drivers of soil degradation in Germany is climate change – however, its direct and indirect (via adaptation strategies) effects are still only insufficiently understood. With respect to soils and agriculture, it is increasingly obvious that climate change will lead to more and longer drought periods in Central Europe (Samaniego et al., 2018), making adaptive changes to soil management necessary, e.g. shifts in crop rotations (Peichl et al., 2019) and increased reliance on irrigation (Riediger et al., 2016). Different climate adaptation options differ in their effects on soil functions

(Hamidov et al., 2018). Climate change-induced weather extremes could also interact with mechanical pressures to aggravate water and wind erosion (Borrelli et al., 2020, 2017). Rising industrial demand for biomass, associated with the bioeconomy (Bruckner et al., 2019), and shifts in consumer demand in response to increasing sustainability challenges can also be expected to interact with the other drivers of soil degradation.





*Q5: Which stakeholder groups have interests associated with particular drivers (qui bono?)?*

As already mentioned in the discussion of target adequacy, the identification and analysis of stakeholders in the 'soil governance field' (Jürges, 2016) is a major research gap; accordingly, we are not aware of any literature looking at questions of the relationship between vested interests and soil degradation drivers.

### 3.2.3 Instrument adequacy

*Q1: Which instruments address the environmental objective? Are there interdependencies?*

The spectrum of policy instruments that influence agricultural soil management is quite broad, as demonstrated by the comprehensive EU-level compilation in the Ecologic inventory (Frelih-Larsen et al., 2016; see also Ronchi et al., 2019). However, when it comes to policy instruments that address soils *explicitly* and that have a *direct* influence on agricultural soil-relevant decision making and behaviour of relevant actors, member-state level GAEC specifications and AECM appear to be the most important; in the following, we will focus on these two (as noted above, the BBodSchG is a policy document of

mainly declaratory nature). Regarding interactions between instruments, it should be noted here that given the overall structure of the CAP, GAEC-based measures should be viewed as defining the mandatory baseline for agricultural soil management, while AECM attempt at incentivizing voluntary actions that go beyond this baseline. In Table 2, we list all arable-land AECM across German federal states (CAP funding period 2014–2020) that can be directly linked to soil protection via the soil pressures as stated in Techen and Helming (2017). The following information on AECM is based on Deutsche

Vernetzungsstelle Ländlicher Raum (n.d.) and complementary searches on responsible agencies' websites. In most federal states, there are also AECM for extensive management of permanent grassland that affect soil functions – however, these have explicitly biodiversity-oriented goals and are therefore excluded here.

**Table 2 Soil-related AECM in German federal states. Notes: BB – Brandenburg/Berlin, BW – Baden-Württemberg, BY – Bavaria, HE – Hesse, MV – Mecklenburg–Vorpommern, NI – Lower Saxony/Bremen, NW – North Rhine–Westphalia, RP – Rhineland–**
**Palatinate, SH – Schleswig–Holstein/Hamburg, SL – Saarland, SN – Saxony, ST – Saxony-Anhalt, TH – Thuringia**

| AECM/Federal states | BB | BW | BY | HE | MV | NI | NW | RP | SH | SL | SN | ST | TH |
|---|---|---|---|---|---|---|---|---|---|---|---|---|---|
| Diverse crop rotation | x | x | x | x | x | x | x | x | x | | | x | x |
| Cover crops | x | x | x | | | x | x | x | x | x | x | x | |
| Stubbles over winter | | | | | | | | | | | x | | |
| Reduced/no-till | | x | x | | | x | | | | | x | | |
| Erosion protection strips | | | x | x | | x | | | | | | | x |
| Erosion protection hedges | | | | | | x | | | | | | | |
| No herbicides | | x | | | | | | | | | | | |
| Precise fertilization | | x | | | | x | | | | | | | |
| Precision farming | | x | | | | | | | | | | | |



The most basic result of the information collated in table 2 is that two soil-related AECM – diverse crop rotations and cover

crops[5] – are most common, while other AECM addressing soil management are only present in a small subset of federal states

each. Furthermore, there are no result-based or otherwise site-specific soil-related AECM – in fact, this is a general challenge

in the EU as a whole, where result-based schemes are so far restricted to biodiversity protection (Bartkowski et al., 2019).[6]

*Q2: How is the correspondence between the instruments and known drivers?*

In Table 3 we link the identified AECM to soil pressures and soil functions. To do this, we use the results of a systematic

review on agricultural land management and soil properties (Chapman et al., 2018), whose gaps we complement by two other

reviews: one on the effects of diversified crop rotations on soil properties (Bai et al., 2018) and one on the effects of tillage

intensity of soil organic carbon (Haddaway et al., 2017). We combine the information derived from the reviews with a mapping

of soil properties and soil functions (Vogel et al., 2019). We focus on those four soil functions identified by Vogel et al. (2019,

2018) that exhibit public good characteristics, i.e. we ignore biomass production/fertility. The four considered soil functions

are: nutrient cycling (NC in Table 4), carbon storage (CS), water storage and filtration (WSF) and biological diversity (BD).

**Table 3 German soil-related AECM and their relation to soil pressures and soil functions. Notes: Soil pressures based on Techen and Helming (2017); soil functions: NC – nutrient cycling, CS – carbon storage, WSF – water storage and filtration, BD – biodiversity; x – positive effect on property strongly associated with soil function according to Vogel et al. (2019), (x) – positive effect on properties with minor association to soil function, 0 – no effect on properties associated with soil function, - – negative effect on properties associated with soil function, ? – effect not known.**

| AECM | Soil pressures addressed | Soil functions addressed | | | |
|---|---|---|---|---|---|
| | | NC | CS | WSF | BD |
| Diverse crop rotation | Crop rotations | (x) | x | (x) | (x) |
| Cover crops | Crop rotations, mechanical pressures | ? | 0 | - | ? |
| Stubbles over winter | Crop rotations, mechanical pressures | ? | 0 | 0 | (x) |
| Reduced/no-till | Mechanical pressures | ? | 0 | (x) | (x) |
| Erosion protection strips | Mechanical pressures, spatial patterns | ? | ? | ? | ? |
| Erosion protection hedges | Spatial patterns | (x) | x | x | (x) |
| No herbicides | Inputs into soil | ? | ? | ? | ? |
| Precise fertilization | Inputs into soil | ? | ? | ? | ? |
| Precision farming | Inputs into soil | ? | ? | ? | ? |

As shown in Table 3, the effects of most soil-related AECM on soil functions are either zero or unknown (including for the

widespread cover crops), exceptions being reduced tillage and hedge planting, where the literature suggests positive effects

---

[5] Note that cover crops are also one of the Ecological Focus Areas (EFAs) options for fulfilling the so-called greening requirements for CAP direct payments; in fact, they are pretty often chosen (Zinngrebe et al., 2017). Given that greening is usually justified in terms of biodiversity conservation, however, we restrict ourselves to pointing this out.

[6] The only result-based agri-environmental payment scheme focusing on soils that we are aware of is the pilot 'Klimaschutz durch Humusaufbau' programme in the Swiss canton Basel-Landschaft, which started in 2019.




across soil functions. This is in line with Vrebos et al. (2017), who showed on the EU level that soil functions are addressed by policies in a rather incoherent and nonsystematic manner.

Looking at the soil pressures associated with the AECM, most measures address impacts related to (narrow) crop rotations and
mechanical pressures, while spatial patterns and inputs into soil are seldom addressed. Furthermore, given our focus on CAP-related instruments, other, broader drivers of soil degradation are not addressed. There is a general lack of dedicated instruments addressing the impacts on agricultural soils from industrial and consumer demand, apart from organic food labels. Also, climate change as a driver of soil degradation, but also soil's role in mitigating climate change has not been addressed in a systematic manner by means of policy instruments, despite initiatives such as 4p1000 (Rumpel et al., 2020) or private soil
carbon certification schemes (Wiesmeier et al., 2020). Furthermore, land availability remains an issue, even though options for novel instruments related to land take (Marquard et al., 2020) and rearrangement/reallocation (Bartkowski et al., 2018; Binder, 2019) have been addressed in the scientific literature.

A remarkable example of policy instruments ignoring relevant drivers despite available knowledge and data is discussed by Siebert (2020), who looked at the erosion cadastres used in German federal states to implement GAEC rules. She shows that
in most cases, not all relevant factors affecting soil erosion risk have been included in the definition of risk zones with management restrictions; for instance, rain erodibility was ignored in the Saxonian erosion cadastre, thus leading to an overly optimistic assessment of the extent of areas affected by erosion risk. Moreover, using the case study of bioenergy-relevant crops, Siebert (2020) argues that the common erosion cadastre approach ignores the importance of specific crops in determining erosion risk (see also Borrelli and Panagos, 2020).

*Q3: Are the individual instruments effective and efficient?*

There is a general paucity of studies looking at the effectiveness of policy instruments in improving the condition of soils, especially going beyond analyses of soil erosion and towards the effects on soil multifunctionality. Among the few available pieces of evidence, modelling studies suggest that the introduction of GAEC rules as part of CAP cross-compliance has reduced soil erosion rates (Borrelli and Panagos, 2020), while there is also tentative evidence suggesting that crop diversification may
actually have the opposite effect (Gocht et al., 2017). In general, however, there is an urgent need for more empirical and modelling studies of the effectiveness, as well as investigations into the efficiency (e.g. cost-effectiveness) of soil-related policy instruments on soil functions.

*Q4: Is the mix of instruments coherent? Are there overlaps or gaps?*

Lack of coherence of the soil-related policy mix on the European level has been bemoaned in the literature (Ronchi et al.,
2019), which is related to a lack of an overarching EU policy framework explicitly addressing soils (Glæsner et al., 2014). Table 2 shows clearly the strongly varying number and extent of soil-targeted AECM across German federal states, just as they vary across member states of the EU.

Given that all relevant policy instruments are related to the CAP, they are rather well aligned – for instance, AECM are required to only incentivize practices that are not mandatory under GAEC or (in the case of cover crops) already used to fulfil greening
requirements. Overall, there are few distinct policy instruments related to soils, so one cannot observe any apparent





contradictions or incoherences, unless looking beyond soil-targeting policies and including those with indirect effects (e.g. bioenergy subsidies, see Siebert (2020)). Even incentives for seemingly contradictory practices (e.g. for many soil types and crops, it is not feasible to farm without herbicide application while at the same time applying no-till practices) can be interpreted as reflecting the heterogeneity of soils and business models that need to be addressed by policies. The problem therefore does

not seem to be (primarily) coherence, but rather ambition and effectiveness.

### 3.2.4 Behavioural adequacy

*Q1: How well do the underlying behavioural assumptions of the instruments reflect actual behaviour of the target population?*
As discussed above, the main policies addressing agricultural soil management in Germany are GAEC and AECM, whereby the former secure a minimal baseline of soil protection, while the latter address soil protection and multifunctionality

(somewhat) more broadly and comprehensively. Relying on mandatory instruments, such as GAEC, implies the (tacit) assumption that agents will behave in accordance to them. Meanwhile, empirical evidence suggests that reality may not be as simple, and that in absence of effective monitoring, there are various reasons for farmers to violate mandatory requirements (see Gaymard et al., 2020). Furthermore, the minimal baseline provided by GAEC and also, less concretely, the BBodSchG, means that ultimately, soil protection is mainly based on voluntary, monetary incentives. It is well known that economic

considerations are the strongest determinants of farmers' management decisions, but it has also repeatedly been shown that other motivational factors also play a role (Bartkowski and Bartke, 2018).
A common indicator of success (conditional on behavioural adequacy) of voluntary incentive-based instruments is their uptake. This is strongly dependent on payment levels, but also on goodness-of-fit in relation to the usual practices and operation procedures as well as available equipment of a farm (and, on the other side of the equation, on public budget availability).

Accordingly, uptake rates vary across AECM types. Overall, irrespective of the specific type of AECM (soil- and non-soil-related), uptake levels between 5 and 20% of arable land area are common (uptake in grasslands is usually higher) (Grajewski, 2016). Moreover, there is a strong tendency to adoption of the 'easiest' measures, which limits the effectiveness of AECM (Bartkowski and Bartke, 2018; see also Zinngrebe et al., 2017). However, uptake data alone are a highly imperfect indicator of behavioural adequacy, because of potentially confounding factors (especially budget constraints faced by administrative

bodies). There is a need for more dedicated analyses of behavioural factors that determine AECM uptake.
*Q2: How well do the instruments reflect the heterogeneity of behavioural characteristics of the target population?*
Braito et al. (2020) show for Austria that one can distinguish different groups of farmers whose basic motivations for sustainable soil management differ, ranging from purely economic motives (based on AECM payment rates) to intrinsic motivations that may only require information and training rather than monetary incentives. Neither AECM nor GAEC seem

to reflect the behavioural heterogeneity of farmers – they offer standardized incentives with limited room for adaptation to biophysical and behavioural characteristics.
*Q3: Are there behavioural complementarities or contradictions between the instruments?*



We are not aware of any research into the interplay of mandatory and incentive-based instruments with regard to farmers' environmentally relevant behaviour. In the UK context, it was argued by Blackstock et al. (2020) that availability of advisory
services can be a precondition for AECM uptake (see also Ingram and Mills, 2019). However, given a lack of similar analyses in Germany, as well as the underrepresentation of sustainable soil management in the very heterogeneous German system of agricultural extension services, this cannot be verified in the context of our case study.

## 4 Lessons learned

### 4.1 The case study

This article used the GDF to provide an exploratory analysis of the German agricultural soil governance. For all four dimensions of the framework – target adequacy, object adequacy, instrument adequacy and behavioural adequacy – we found deficits, most of which can be linked to the generally underappreciated role of sustainable soil management in German and EU agri-environmental policy. In fact, soil-related policies are fragmented, underdeveloped and not comprehensive, e.g. they do not explicitly and directly address all soil functions and the interactions between them. This lack of coherence and depth
starts already at the level of political targets and propagates throughout the frameworks four dimensions. We would like to underline the following deficits that appear particularly concerning:

- When looking at explicitly soil-dedicated policies, three major drivers of soil degradation (potential), namely climate change (both in terms of soil's mitigation potential and the interaction between sustainable soil management and climate change adaptation), the bioeconomy as a new source of demand for biomass, and the food chain beyond
production remain largely unaddressed. However, this may change due to the introduction of the European Green Deal (Montanarella and Panagos, 2021).
- Relatedly, the multifunctionality of soils seems largely unaddressed; targets found both in strategic documents and, implicitly, in policy instruments, do not reflect well the multiple functions and ecosystem services provided by soils, nor the interactions (especially: trade-offs) among them.
- The effectiveness of dedicated agricultural soil policy instruments is questionable, which partly reflects their rather rudimentary design (e.g. lack of more site-specific and/or result-based instruments addressing the heterogeneity of soils).
- The heterogeneity of the farmer population and the complex determinants of their behaviour are not well reflected in existing policy documents, likely because of simplistic understanding of farmers' motivation by policy makers (see
Brown et al., 2020).

On the positive side, the combination of a mandatory minimum standard (GAEC) with incentives for the adoption of 'additional' soil protecting management practices (AECM) seems to be a good starting point for further developing the production-side of soil governance in Germany and the EU.



While trying to provide tentative answers to the questions formulated in section 2, we identified the following research gaps,
where evidence is missing and where there is a need for dedicated research:

- There is an urgent need for more research attention to the *actors/stakeholders* involved in and affected by soil governance, their preferences, vested interested, their association with drivers of soil degradation etc.
- The *temporal and spatial dimensions* in the context of targets, drivers, preferences and instruments, including temporal and spatial mismatches, require more research.
- Studies into possible *prioritization* of different drivers of soil degradation would be helpful for more effective and rapid policy formulation.
- There is a general paucity of research into *effectiveness of policy instruments* as well as, relatedly, the *behaviour of relevant actors* (particularly farmers) in the specific context of agricultural soil management.

### 4.2 The GDF applied to agricultural soil policy assessment

Overall, we found the GDF useful in generating a broad overview about agricultural soil governance in Germany. However, we also identified a trade-off between the (realistic) level of detail and the breadth and number of issues addressed – being so comprehensive, the disruptions framework requires the synthesis of very many pieces of evidence, which can be challenging within limits of time and other resources. In our case study, we restricted ourselves to policy documents and instruments that address agricultural soils *explicitly*, knowing that one could include many more policies that have a more implicit or indirect
yet non-negligible relevance in the analysed governance context. Moreover, there is the challenge of navigating between different levels of analysis – while policy targets are often formulated at national and supranational levels, specific policy instruments are often designed and implemented at national or even, as in the case of Germany's agri-environmental policy, regional levels. At the same time, the GDF has proven quite useful in facilitating the structured identification of research gaps relevant to a comprehensive analysis of natural resource governance.

**Author contribution**

All authors conceptualized the paper. BB prepared the manuscript with contributions from all co-authors.

**Competing interests**

The authors declare that they have no conflict of interest.





**Acknowledgements**

We would like to thank Nele Lienhoop and Sylvia Bittner for making available their overview about soil-related AECM; Marieke Baaken for suggesting relevant meta-analyses to complement Table 3; and Ulrich Weller for consultation on matching AECM with soil functions. This work was funded by the German Federal Ministry of Education and Research (BMBF) in the framework of the funding measure 'Soil as a Sustainable Resource for the Bioeconomy – BonaRes', project 'BonaRes (Module B): BonaRes Centre for Soil Research, subproject A' (grant 031B0511A).

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
