# Peer review of "Application of the Governance Disruptions Framework to German agricultural soil policy"

_SOIL, 2021_

## Referee Comment (RC1)

[referee-annotated manuscript omitted]

---

## Author Response (AR1)

Dear editor, dear reviewers,

since the revisions described here are based on our comments already made in the discussion forum, here we only provide concise descriptions of what and where we changed in the manuscript to accommodate the reviewers' comments. Note that we also slightly rephrased Q1 and Q3 in section 2.1 on target adequacy, in order to improve their applicability and so that they fit better their application in section 3.2.

| Comment | Response |
|---|---|
| Reviewer 1 | |
| The subject is interesting and the various aspects of soil protection are covered. However, the paper fails to address one major problem of all legislative attempts to apply a soil protection law which is the lack of data. It is not very useful to invoke to "explicitly and directly address all soil functions and the interactions between them" when science is far from attain such a target or even defining it. More space should be given to the root problem of the scarcity and quality of data which is, after all, the very reason why soil is neglected in the legislations. The difficulties in obtaining the data and their variability should be taken into account and duly commented upon. | We addressed the issue of data availability in section 3.2.1 on target adequacy by adding the following passage: "These gaps may be a reflection of data (non-)availability as an important constraint of environmental policy, though the recently launched EU Soil Observatory has the potential to improve the situation significantly (Montanarella and Panagos, 2021). In this sense, soil policy can be considered an art of making decisions and creating instruments in the absence of perfect knowledge and in a way that allows to adapt to new knowledge." (lines 193–7 in the revised, track-change document)

Also, in section 3.2.4, we again address the more specific context of result-based payments: "However, this [lack of result-based soil schemes] may reflect the challenges in terms of monitoring 'results' such as soil functions (Jeffery and Verheijen, 2020; Vogel et al., 2019), whereas innovative, e.g. model-based payment schemes might be a promising alternative (Bartkowski, 2021; Bartkowski et al., 2021)." (lines 302–4) |
| Reviewer 2 | |
| The article is based on very specific theoretical concepts (difficult to grasp for non-specialists) and a proper understanding of these concepts would require reading also the numerous publications cited (e.g. p. 111 so-called Tinbergen rule, p. 203-206 even if consulting the cited literature, it is not clear why organic fertilization may be detrimental to water quality). A more tangible approach, with examples, would increase the impact of the article. | We deleted the (unnecessary) reference to the Tinbergen rule and simplified the point made (lines 111–2 in the revised, track-change document). Also, we specified the organic fertilizer point by adding the following half-sentence: "as exemplified by the high spatial correlation of livestock production (and thus local availability of organic fertilizer) and nitrate pollution in North-Western Germany" (lines 211–12) |
| From a pragmatic policy making point of view, the conclusions of the article (4. Lesson learned) do not | In addition to the small insertion on the importance of communication for uptake of AECM in section 3.2.4 (see below), we added the following to section |

sufficiently reflect the importance of the analysis performed and the recommendations lack relevance or are a little bit weak ("urgent need for more research").

A proposal would be to develop reflection/recommendations on how to include the following topics in policy frameworks, possibly with prioritization:

1. Monitoring. The first question (target adequacy; specific environmental objectives, concrete indicators) clearly suggests the importance of setting targets and measuring their achievement. Even if this point is obvious, it should be mentioned here.
2. Behavioural changes. The original aspect of the GDF is to raise the importance of individual and societal behaviours for the implementation of conservation/protection measures of natural resources. This is now widely accepted in economics (behavioral economics), but has not been sufficiently taken into account in policy designing.
3. Communication. The issue of (lack of) communication is well known, and also underlined by the authors (p. 387). The transfer of knowledge (i.e. of appropriate and comprehensible information between different levels and stakeholders) constitutes a major challenge and a critical phase in soil protection and policy design. It might be argued that this point is not

4.1 (lines 422–31): "Against this background, two major practical consequences for soil policy appear particularly salient: first, effective soil policy requires clearly and realistically formulated targets that take into account the current understanding of underlying mechanisms and availability of data for monitoring the success of policy interventions. Second, much knowledge is available about behavioural factors that affect the effectiveness of environmental policies, including soil-related management (Bartkowski and Bartke, 2018). However, the (implicit) assumptions reflected in conventional policy design are rather simplistic (Brown et al., 2021). This calls for more consideration of behavioural factors in soil policy design. In addition to these two issues, and given that soil policy is covered at various governance levels (from EU to federal states and further down), while being implemented 'on the ground' by farmers, communication (of and about soil policy targets, sustainable management practices, legal competencies, administrative rules etc.) across governance levels and among stakeholders is crucial for successful soil protection."

Furthermore, in section 4.2, we expanded the final statement, which now reads as follows: "One may say that in this respect, the GDF reflects the challenges of the policy arena we have applied it to, where communication across levels is essential (see above). At the same time, the GDF has proven quite useful in facilitating the structured identification of research gaps relevant to a comprehensive analysis of natural resource governance. Following this exploratory application, the GDF can now be tailored to more specific aims and contexts, in order to illuminate particular aspects of the natural resource governance. This may include pragmatic simplification to facilitate GDF's use as an analysis tool for policy makers." (lines 454–9)

| | |
|---|---|
| sufficiently and explicitly addressed in the GDF. | |
| Finally, an adaptation (and simplification) of the GDF to the field of soil legislation would be a valuable tool in the policymaking. | Regarding the adaptation of the GDF to the realities of soil legislation processes, we concur that this could be helpful – our paper's aim has been to demonstrate how the GDF in its "complex" form can be applied, and simplifying it for specific purposes would be a next step. We added this to the conclusion section (see above). |
| p. 192-193 SDG indicators: 15.3.1 Proportion of land that is degraded over total land area includes explicitly the soil organic carbon stock. | Corrected (lines 192–3). |
| table 3 Possibly revise the table: cover crops (may) have an effect on CS, herbicides on BD | We added the reviews by Rose et al (2016) and Gunstone et al (2021) and revised the herbicide effect on BD in Table 3. |
| p. 359-361 As the § addresses the behavioural adequacy, "other motivational factors" should be more detailed. | We added the following sentence in lines 372–4: "For instance, Bartkowski and Bartke (2018) show that, depending on the specific context, factors such as general pro-environmental attitudes or problem perception can play an important role in soil-related decisions."

Also, in lines 378–80, "Also, it has been shown in other contexts that transparent communication of AECM goals, administrative rules and responsibilities as well as perceived administrative effort associated with participation are important for uptake (Brouwer et al., 2015; Mack et al., 2020, 2019)." (addressing the communication point from above) |

---

## Author Response (AR2)

Dear Editor and reviewer,

thank you for the helpful and constructive comments. Below we respond to each comment while referring to the revised manuscript:

| Comment | Response |
| --- | --- |
| it is not completely clear how they are combined (technically) [line 55] | Thank you for pointing this out! The combination is rather context-specific and qualitative; we've added a short explanation, so that the sentence now reads "Therefore, the benefit is in looking at all four of them together and exploring interlinkages (e.g. disruptions in one dimension that affect another dimension; or related disruptions across dimensions)." (lines 55–6 in the track-change document) |
| This also manifests itself in footnote 1 (p. 4): The footnote seems to be quite difficult to understand, if the potential reader is not so deep into the topic (relation between GDF and DPSIR model). | We have removed the reference to DPSIR in the footnote in order to avoid confusion. |
| The withdrawal of the Soil Framework Directive happened to be in May 2014, not in 2006 [line 25]. The reason for withdrawal (GER): claimed to be too expensive (but also: difficult situation in discussions on policy level/ with stakeholder groups (farmers` association)) [line 25f.]. | Thank you! We've corrected the withdrawal year. |
| For considered soil functions (based on Techen and Helming 2017), maybe you could hint on the EU H2020 project Land Management Assessment Research Knowledge base (LANDMARK). LANDMARK's classification is quite similar (http://www.soilnavigator.eu/). | Thank you for this suggestion; we now refer to the LANDMARK approach in line 181, in the context of soil functions as defined in the BBodSchG. |
| Maybe, you could include further BONARES's outcomes when mentioning surveys on stakeholders' opinions and literature research on governance styles, i.e. Ledermüller/ Fick/ Jacobs 2021 and Marx/ Jacobs 2020 | We added a reference to Marx & Jacobs in lines 186–7 as another example of target inadequacy. |
| The critique on the concept of 'Good agricultural practice' being imprecise, is certainly justified [line 60]. Maybe, you could also point out that result-based approaches do fit better to the EGD targets (and its sub-strategies) [line 63f.]. | We added a reference to the EGD's call (formulated e.g. in the F2F strategy) for a more result-based approach in lines 64–5. |
| Similarly revealing is the classification of 'soft', informal instruments into the context of policy instruments and stakeholders' response [line 108/ 135]. What is missed | We added a brief reference to social influences in line 136. |

| | |
|---|---|
| though is that other motivational factors (e.g. comparison with the neighbour farmer) also play a role [line 360]. | |
| Maybe, an outlook on reaching EGD/ EU FTF targets would be nice, i.e. If a better governance level would be reached (by the GDF tool), the results/ site-specific efforts of each farmer could be (more easily) assessed and CAP payments may be combined with these efforts for sustainable soil management on plot level in the nearer future. | Thank you for the suggestion; as we consider our analysis rather exploratory, we would prefer to not draw too strong policy-relevant conclusions, especially given that the EGD hasn't been the main focus on the analysis. |
| line 110: Delete the word 'which' and insert the word 'that'. | Done. |
| line 112: Delete the word 'that'. | Done. |
| line 176/ 187/ 300: Check punctuation in connection with footnotes. | Checked, it's all according to English punctuation norms (footnote after punctuation). |
| line 313/ Table 3: the values (x), x, ?, - and 0 seem to be difficult to reckon visually. If you choose '-' for a negative effect, a positive effect should me marked by '+' (and not by x). In Figure 1, colours are used, so I suggest using colours here as well to support your assessment. | Thank you for this suggestion! We changed "x" to "+" and added some minimal colour coding (green for +, light green for (+), orange for -); however, we are not sure whether this is in line with the journal graphical standards. |
| line 357/ 358: Which soil types do you mean? Soil types characterized by a high percentage of clay/ clayey soils? Which crops do you mean? Maize, winter wheat? (a short example each might be helpful) | We are not aware of any systematic investigations into this; our formulation is based on anecdotal evidence and reports from farmers. We therefore rephrased and now say "in many contexts", rather than "for many soil types and crops". Furthermore, we added a reference on the adoption of no-till practices in organic agriculture in Germany. |
| line 358: The expression 'not feasible' might be too strong. There are efforts on how to farm no-till without herbicide application, but farmers must be well educated/ have excellent knowledge to do this. | We rephrased as "currently not feasible". |